# Three dimensional band-filling control of complex oxides triggered by interfacial electron transfer

Meng Meng[1], Yuanwei Sun[2], Yuehui Li[2], Qichang An[1,3], Zhenzhen Wang[1,3], Zijian Lin[1,3], Fang Yang[1], Xuetao Zhu[1,3,4], Peng Gao [2,5✉] & Jiandong Guo [1,3,4,6✉]

The *d*-band-filling of transition metals in complex oxides plays an essential role in determining their structural, electronic and magnetic properties. Traditionally, at the oxide heterointerface, band-filling control has been achieved via electrostatic modification in the structure of field-effect transistors or electron transfer, which is limited to the quasi-two-dimension at the interface. Here we report a three-dimensional (3D) band-filling control by changing the local lattice coordination in a designed oxide heterostructure. At the $LaCoO_3/LaTiO_3$ heterointerface, due to the Fermi level mismatch, electrons transfer from $LaTiO_3$ to $LaCoO_3$. This triggers destabilisation of the $CoO_6$ octahedrons, i.e. the formation of lattice configurations with a reduced Co valence. The associated oxygen migration results in the 3D topotactic phase transition of $LaCoO_3$. Tuned by the thickness of $LaTiO_3$, different crystalline phases and band-fillings of Co occur, leading to the emergence of different magnetic ground states.

[1] Beijing National Laboratory for Condensed Matter Physics and Institute of Physics, Chinese Academy of Sciences, Beijing, China. [2] International Center for Quantum Materials, and Electron Microscopy Laboratory, School of Physics, Peking University, Beijing, China. [3] School of Physical Sciences, University of Chinese Academy of Sciences, Beijing, China. [4] Songshan Lake Materials Laboratory, Dongguan, Guangdong, China. [5] Collaborative Innovation Center of Quantum Matter, Beijing, China. [6] Beijing Academy of Quantum Information Sciences, Beijing, China. ✉email: p-gao@pku.edu.cn; jdguo@iphy.ac.cn

Complex oxides display an unprecedented diversity of fascinating functionalities arising from the strong interactions between their charge, spin, orbital and lattice degrees of freedom[1]. Besides the non-isovalent chemical doping, the $d$-band-filling control can be achieved at oxide interfaces by the electric field effect. Various intriguing phenomena have been observed, e.g. insulator-superconductor transition[2], metal-insulator transition[3] and magnetoelectric effects[4,5]. Owing to the potential discontinuity at complex oxide heterointerfaces, marked charge modification might occur, leading to electronic reconstructions and further the emergent electronic and magnetic states[6–16]. However, modification via either the electric field effect or interfacial charge transfer only works within a confined quasi-two-dimensional region near the interface due to the short electrostatic screening length of complex oxides[17]. Recent advance in achieving the three-dimensional (3D) $d$-band-filling control is through tuning the oxygen coordination number of the transition metal ions[18–20], limited to the oxides that adopt the ordered oxygen-deficient structures as the thermodynamically stable phase.

Among the $3d$ transition metal oxides, the most remarkable property distinguishing cobalt oxides from the others is that Co ions can accommodate various spin states dictated by the relative strength of the crystal field splitting $\Delta_{cf}$ and intra-atomic (Hund's) exchange energy $\Delta_{ex}$. For example, octahedrally coordinated $Co^{3+}$ ions in $LaCoO_3$ (LCO) could have three spin states (Fig. 1a–c): low-spin (LS, $S=0$) $t_{2g}^6$ (where $\Delta_{cf}$ is slightly larger than $\Delta_{ex}$), intermediate-spin (IS, $S=1$) $t_{2g}^5 e_g^1$, and high-spin (HS, $S=2$) $t_{2g}^4 e_g^2$ (where $\Delta_{ex}$ is barely larger than $\Delta_{cf}$)[21–23], while the selection decided by Hund's rule states that $Co^{2+}$ always adopts the HS

$t_{2g}^5 e_g^2$ state (Fig. 1d). The combination of the valence and spin states determines the electrical and magnetic properties of LCO. Bulk LCO is a nonmagnetic insulator owing to the LS state $Co^{3+}$ ions[23,24]. As $\Delta_{cf}$ is very sensitive to the Co–O bond length and angle[25], the IS or HS state of $Co^{3+}$ can be stabilised by the in-plane tensile strain in LCO thin films, resulting in a ferromagnetic (FM) insulating ground state[25–30]. It was reported that two metastable, anion-ordered phases were prepared by the special 'kinetically controlled' reduction of LCO so that the Co $d$-band-filling could be modified between $3d^6$ and $3d^7$ as the lattice coordination varied[31,32]. However, it is still challenging to tuning the band filling by design, since the perovskite LCO is the most thermodynamically stable crystalline phase[33].

In this work, we report the 3D modulation of the $d$-band-filling and topotactic phase transition of LCO by designing LCO/LaTiO$_3$ (LTO) heterostructures (Fig. 1e). Due to the significant Fermi level ($E_F$) mismatch at the heterointerface, electrons transfer from LTO to LCO, which drives the perovskite LCO instability. Oxygen ions are then released from LCO to form local lattice configurations with a reduced Co valence state. The redistribution of these local configurations results in the 3D topotactic phase transition throughout the LCO film. The reduction of LCO can be tuned by the thickness of LTO, leading to a magnetic transition from FM to antiferromagnetic (AFM) ordering. Our findings provide a simple way to control the electronic configuration for the exploration of 3D emergent properties of complex oxides by designing heterointerfaces. Such 3D modification breaks the limitation of the electric field effect, which only works with an ultra-thin oxide film, leading to potential applications in neuromorphic computing and iontronics.

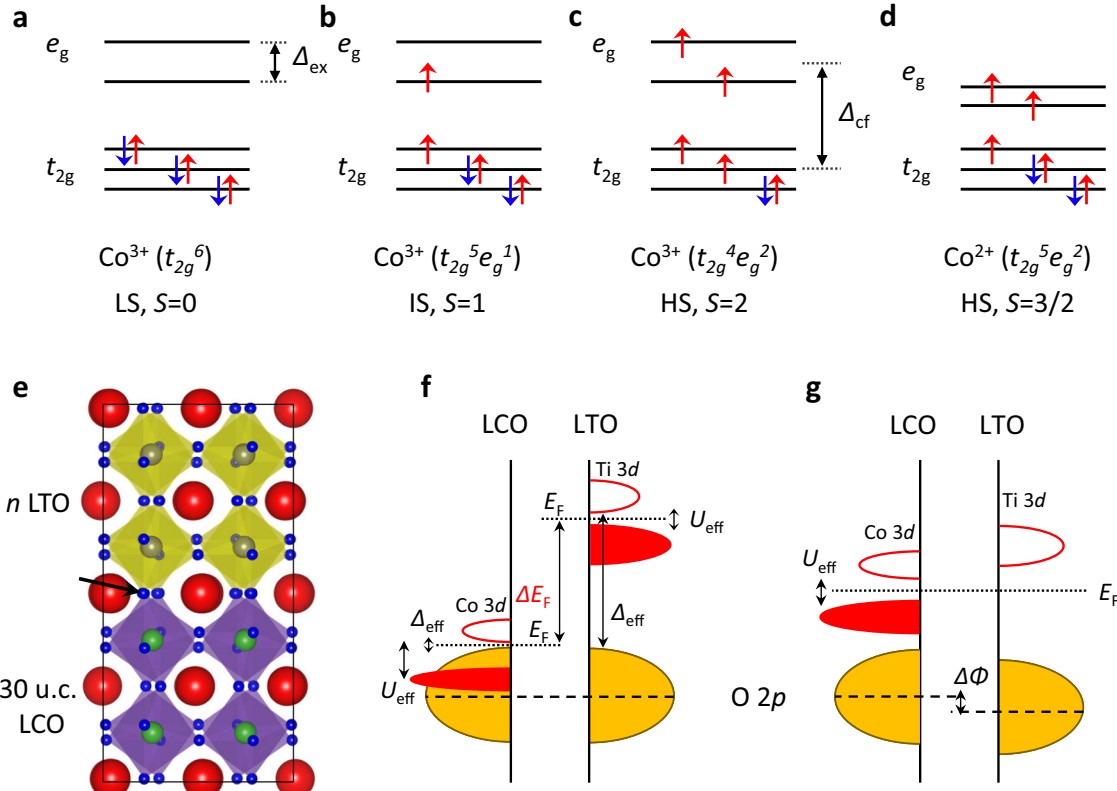

**Fig. 1 Spin states of Co ions and electronic band structure of LCO/LTO. a–d** Spin states of Co ions. $\Delta_{cf}$, crystal field splitting, $\Delta_{ex}$, intra-atomic exchange energy. **e** Schematic diagram of 30u.c.-LaCoO$_3$ (LCO)/$n$-LaTiO$_3$ (LTO) heterostructure grown on SrTiO$_3$ (STO) substrate. We refer to this heterostructure as C30/T$n$. The black arrow indicates the shared oxygen sites at the interface. **f** Electronic band structures of LCO and LTO. $E_F$, Fermi level; $\Delta_{eff}$, charge-transfer gap; $U_{eff}$, Mott–Hubbard gap. **g** The final equilibrium state wherein the $\Delta E_F$ can be removed by charge transfer from Ti to Co. $\Delta\Phi$, electrostatic potential drop.

## Results

The interfaces of complex oxides have been proposed to follow different band-alignment rules due to the invalidity of the single work–function approximation[34]. As schematically illustrated in Fig. 1e, f, since the corner oxygen sites are shared at the interface, the oxygen $2p$ states should be aligned[9–13]. The difference in the charge-transfer gap $\Delta_{eff}$, which is between the filled O $2p$ and unoccupied upper-Hubbard $3d$ states[35], determines the $E_F$ mismatch ($\Delta E_F$) across the interface. For LCO ($3d^6$), it is a narrow gap charge-transfer insulator with $\Delta_{eff} \sim 0.1$ eV[36], while in a prototypical Mott–Hubbard insulator, LTO ($3d^1$) has small Hubbard splitting $U_{eff} \sim 0.2$ eV and large $\Delta_{eff} \sim 4.5$ eV (its $3d$ band is located far above its oxygen $2p$ band)[35]. Therefore, there is a huge $\Delta E_F$ (~4.4 eV) at the LCO/LTO interface, resulting in electron transfer from the lower-Hubbard Ti $3d$ band of LTO to the empty upper-Hubbard Co $3d$ band of LCO. Such electronic reconstruction causes a rigid $3d$ band shift, i.e. the elevation of the Co $3d$ band and the lowering of the Ti $3d$ band, as illustrated in Fig. 1g. Note that the above scenario is confined within the 2D region at the interface, and the modification of the band-filling can be described as $Co^{3+}$ ($3d^6$) + $Ti^{3+}$ ($3d^1$) → $Co^{2+}$ ($3d^7$) + $Ti^{4+}$ ($3d^0$).

The LCO (30 u.c.)/LTO ($n$) (referred to as C30/T$n$ in the following, u.c. = unit cell) heterostructures were grown on TiO$_2$-terminated SrTiO$_3$ (STO) substrates by pulsed laser deposition (see 'Methods' section). We first explore the charge transfer across the LCO/LTO interface via electron energy loss spectroscopy (EELS). Figure 2a shows the line-by-line Co $L$-edge spectra from the LCO/LTO interface (uppermost curve) to the LCO/STO substrate interface (bottom curve) of a C30/T15 heterostructure. Surprisingly, we do not observe any clear peak shifts at different positions, against the interfacial charge-transfer picture. Together with the roughly unchanged Co $L_{3,2}$ ratio (Fig. 2b), we conclude that the valence state of Co in the C30/T15 is spatially homogeneous. However, upon comparing it with the LCO film without LTO (C30/T0), the Co $L_3$ peak of the C30/T15 shifts towards lower energy (Fig. 2c), while the peak position of the La $M$-edge is identical, indicating the decrease of the valence state of Co ions in the C30/T15. Figure 2e shows the Co $L$-edge X-ray absorption spectroscopy (XAS) results. The curve of C30/T15 has split $L_3$ peaks, which shift to lower energy compared with that from the C30/T0 sample. The peak shift and $L_3$ peak splitting are the signatures of the $Co^{2+}$ valence state[15,16]. The difference of the averaged O $K$-edge spectra (Fig. 2d) of the LCO in the C30/T15 and C30/T0 also indicates charge redistribution. There is a pre-peak feature in C30/T0 originating from the excitation of O $1s$–O $2p$ on the O $2p$ holes created by the hybridisation of O $2p$ with Co $3d$[37,38]. However, such a pre-peak is absent in the C30/T15, since there are no O $2p$ holes at the $Co^{2+}$ sites, as previously revealed in CoO[39] and proton-inserted HSrCoO$_{2.5}$[18]. The charge redistribution could also be verified via XAS measurements of the Ti $L$-edge (Fig. 2f). The features exhibit excellent agreement with those of the $Ti^{4+}$ charge state and are remarkably different from the spectra of $Ti^{3+}$, consistent with calculated Ti oxidation states in the LTO layer from the spatially resolved STEM-EELS (Supplementary Fig. 2), demonstrating electronic reconstruction on the Ti site, e.g. Ti $3d^1$ to $3d^0$. The expected electron transfer from Ti to Co indeed occurred in the C30/T15; however, the length scale breaks the 2D limit, reaching the 3D scale.

This 3D charge redistribution affects the lattice structure, which could be observed macroscopically. The bulk LTO has a GdFeO$_3$ orthorhombic structure with pseudocubic lattice parameter $a_{pc} = 3.969$ Å[40]; while the structure of LCO is rhombohedral with pseudocubic lattice parameter $a_{pc} = 3.826$ Å (Fig. 3a)[41]. Figure 3b shows the X-ray diffraction (XRD) $\theta$–$2\theta$ scan patterns of the LCO/LTO grown on STO ($a = 3.905$ Å) substrates.

Without the LTO (C30/T0), only diffraction peaks from LCO (besides those of the STO substrates) were observed (black line). The presence of clear Kiessig fringes indicates the excellent crystalline quality of the film. For the C30/T10 (magenta line) and C30/T15 (light blue line) heterostructures, topotactic phase transitions of LCO occur by interfacing with LTO. The phase in the C30/T10 is consistent with the La$_3$Co$_3$O$_8$ phase, which has a monoclinic u.c. with $a_m = 5.559$ Å, $b_m = 5.415$ Å and $c_m = 11.773$ Å (space group $P2_1$) (Fig. 3a)[32]. Each u.c. consists of double CoO$_6$ octahedral layers and single CoO$_4$ tetrahedral layers along its $c_m$ axis, which could be represented as pseudotetragonal with $a_t = 3.879$ Å and $c_t = c_m/3 = 3.925$ Å. The emergent phase in the C30/T15 is identified as brownmillerite (BM) La$_2$Co$_2$O$_5$, which adopts a distorted orthorhombic u.c. (space group $Pnma$), with $a_o = 5.445$ Å, $b_o = 15.869$ Å and $c_o = 5.692$ Å (Fig. 3a)[31]. It could also be considered pseudotetragonal, with $a_t = 3.937$ Å and $c_t = b_o/4 = 3.967$ Å. In La$_3$Co$_3$O$_8$ or La$_2$Co$_2$O$_5$, the CoO$_6$ octahedral and CoO$_4$ tetrahedral sub-layers are stacked alternatively with long-range order along the $c_t$ axis, giving rise to the (0 0 $k/3$) or (0 0 $k/2$) fractional diffraction peaks (indices with respect to the average perovskite u.c.) as shown in Fig. 3b.

The structural distortion induced by the inserted CoO$_4$ tetrahedral layer could be visualised directly in the real space via scanning transmission electron microscopy (STEM). Figure 3c displays a high-angle annular dark-field (HAADF)-STEM image of the C30/T15, taken along the STO [110] zone axis. The most straightforward observation is the lateral superstructure (parallel to the interface) with doubled periodicity, which could be seen as a signature of the BM phase with long-range alternative stacking of the octahedral and tetrahedral sub-layers[19]. We also performed a STEM simulation based on the BM La$_2$Co$_2$O$_5$. The consistency of the experimental and simulated STEM images seen along the $c_o$ axis (inset of Fig. 3c) further confirms the existence of a periodically arranged CoO$_4$ layer. Such a superstructure is absent in the C30/T0 sample (Supplementary Fig. 3), consistent with the macroscopic structural characterisation of XRD.

The reciprocal space map reveals that throughout the topotactic phase transitions, La$_2$Co$_2$O$_5$ is still coherently strained on STO (Supplementary Fig. 4). The out-of-plane lattice parameter $c_t$ and the volume of the pseudotetragonal u.c. show a monotonous increase as the LTO thickness increases, consistent with the monotonous rise of the proportion of $Co^{2+}$ ions associated with the insertion of CoO$_4$ tetrahedral layers into the structure. Quantitatively, it should be noted that the ratio of CoO$_6$ to CoO$_4$ layers in the emerging superstructures equals $30/n$ in C30/T10 and C30/T15 cases, i.e. one u.c. LTO corresponds to the insertion of one tetrahedral layer into the octahedral matrix.

The in-plane lattice parameters of bulk La$_2$Co$_2$O$_5$ and LTO are larger than those of the STO substrate ($a_{STO} \sim 3.905$ Å, $a_{La2Co2O5} \sim 3.937$ Å and $a_{LTO} \sim 3.969$ Å), indicating that the C30/T15 is subjected to compressive strain. From the STEM images, we measured the in-plane lattice parameters of the La$_2$Co$_2$O$_5$ and LTO blocks as $3.90 \pm 0.05$ Å (Supplementary Fig. 5), in accordance with the reciprocal space map. The out-of-plane La–La atomic distance varies between the CoO$_6$ (~3.67 ± 0.1 Å) and CoO$_4$ (~4.32 ± 0.1 Å) sub-layers, as shown in Fig. 3d. Such an interspacing modulation is attenuated towards the LCO/LTO interface and might be associated with the slight intermixing across the interface (Supplementary Fig. 6). Importantly, we observed that the LTO films in both the C30/T10 and C30/T15 maintained a well-defined perovskite structure without forming ordered LaTiO$_{3+\delta}$ phases[42].

Next, we address the 3D modulation of the physical properties of the C30/T$n$. As shown in Fig. 4a, the highly insulating behaviour and the thermally activated character of all heterostructures are clearly observed, with the sheet resistance in the order of

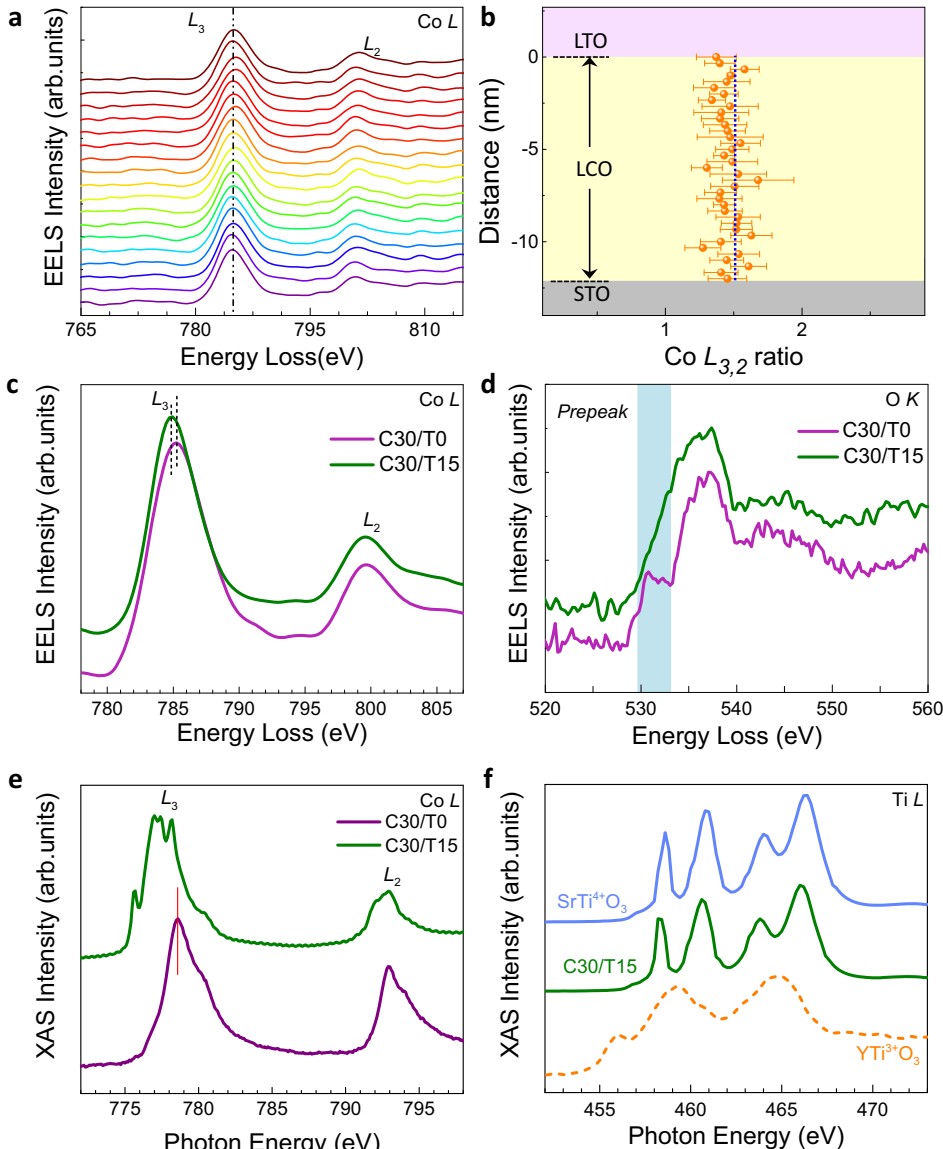

**Fig. 2 EELS and XAS of C30/Tn heterostructures. a** The Co L-edge EELS spectra of C30/T15 at different positions from STO/LCO to the LCO/LTO interface. The bottom curve is from the STO substrate/LCO interface, and the uppermost curve is from the LCO/LTO interface. **b** The corresponding Co $L_{3,2}$ intensity ratio. The LCO/LTO interface sets for zero. The averaged **c** Co L-edge and **d** O K-edge EELS spectra of C30/T15 and C30/T0. The shadow area in **d** indicates the distinct difference in the pre-peak feature. The spectra were averaged from the line scan in the 30u.c. LCO block. **e** Co L-edge XAS spectra of the C30/T15 and C30/T0 samples. **f** The Ti L-edge XAS spectra of the STO substrate, C30/T15, and 100-nm YTiO₃ film from ref. [10].

$\sim 10^9$ Ω/□ at room temperature. The sample with a thicker LTO has a higher content of $Co^{2+}$ ions and larger resistance since electrons are easily localised on the $Co^{2+}$ sites. By fitting the transport data (inset of Fig. 4a), we calculated the activation gap as ~0.32 ± 0.02 eV for the C30/T0, with a monotonous increase as the LTO thickness increases.

There is a substantial change of the magnetic ground states associated with the topotactic phase transitions, as shown in Fig. 4b, c. The tensile-strained LCO film (C30/T0) has the FM ground state. However, for the C30/T10 ($La_3Co_3O_8$) and C30/T15 ($La_2Co_2O_5$), the FM ordering vanishes without any indication of magnetic transition. Considering the spin state and charge state of Co ions (Fig. 1a–d), in the C30/T0, $Co^{3+}$ ions could adopt either the LS or HS state: an LS/HS mixture state[28]. The FM insulating ground state is established via the superexchange interaction within the HS–LS–HS $Co^{3+}$ configuration according to the so-called Goodenough–Kanamori–Anderson rules[43].

However, this interaction is weak and easily vanishes when the HS $Co^{2+}$ ions reach a threshold ratio of ~12.5%[30], since the exchange coupling between two $Co^{2+}$ with a half-filled $e_g$ orbital turns out to be strong and AFM. This explains the disappearance of FM ordering in the $La_3Co_3O_8$ and $La_2Co_2O_5$: they are actually in an AFM ground state as in their bulk form[31,32].

## Discussion

The LCO/LTO interfacial electron transfer has been expected in 2D[9–15]. Surprisingly, however, 3D modulation of the band-filling mediated by the topotactic phase transitions is observed. To determine the mechanism, we performed first-principle calculations on the structural stability of LCO (Supplementary Note 2). The results reveal that, with the extra transferred electrons, the perovskite structure of LCO becomes unstable, while the reduced $CoO_4$ tetrahedral lattice configuration with released $O^{2-}$ ions

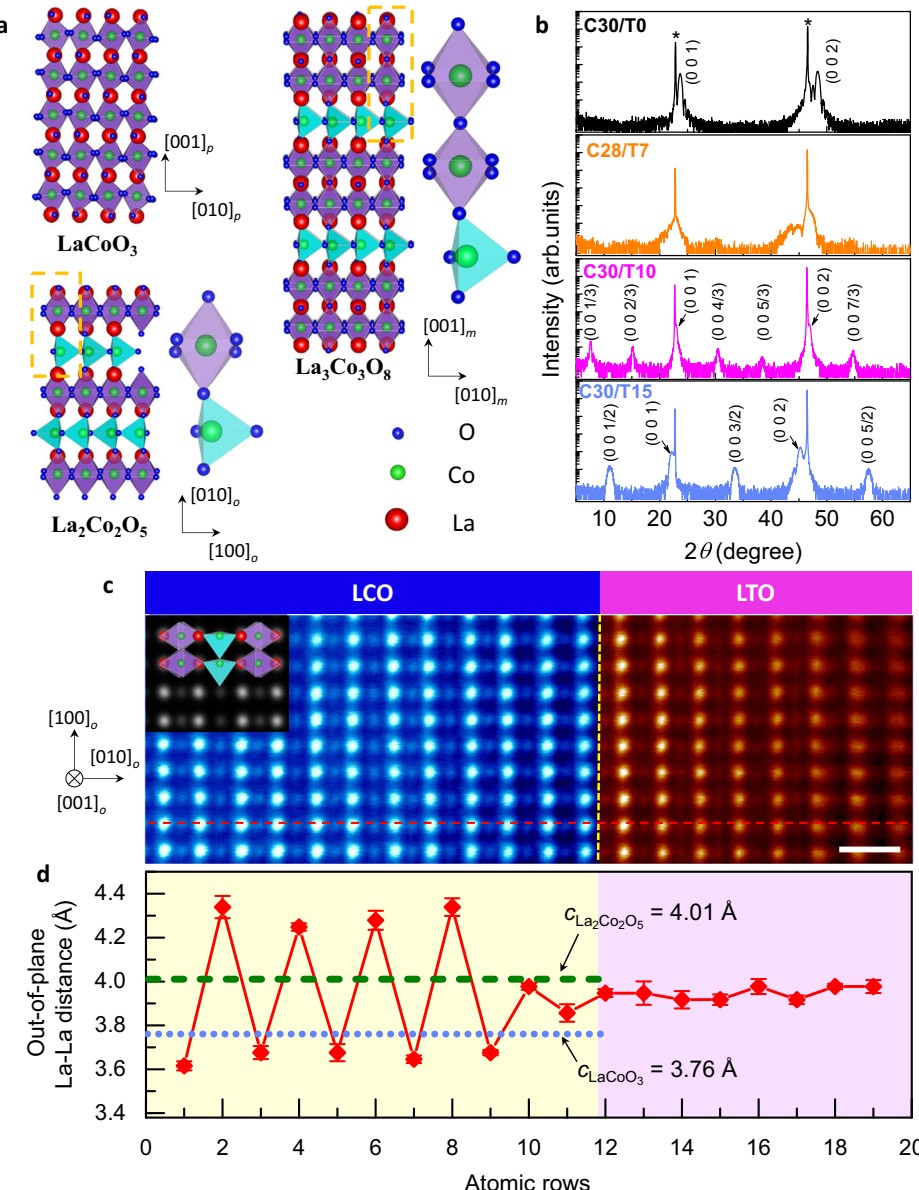

**Fig. 3 Topotactic phase transition in C30/T*n*. a** Schematic crystal structure of perovskite LaCoO₃, La₃Co₃O₈, and brownmillerite La₂Co₂O₅, respectively. Enlarged details are presented to show the stacking sequence of $CoO_6$ octahedrons and $CoO_4$ tetrahedrons along the out-of-plane direction. **b** X-ray diffraction data of LCO/LTO heterostructures. Blackline, C30/T0; orange line, C28/T7; magenta line, C30/T10; light blue line, C30/T15. All indices correspond to the averaged perovskite u.c. Peaks from STO substrates are denoted by asterisks. **c** High-resolution high-angle annular dark-field scanning transmission electron microscopy (HAADF-STEM) image of C30/T15 taken along the STO [110] direction. The scale bar corresponds to 0.5 nm. The inset image shows a simulated STEM image viewed along the $[001]_o$ direction based on the structure of brownmillerite La₂Co₂O₅. The yellow dashed line indicates the LCO/LTO interface, while the red dashed line shows the region used to calculate La–La interatomic distances. **d** Out-of-plane La–La distance along the red dashed line shown in **c**. The error bar shows the standard deviations of the averaged measurements. The cyan dotted line and green dashed line indicates the out-of-plane lattice parameter of LaCoO₃ (C30/T0) and La₂Co₂O₅ (C30/T15) obtained from **b**, respectively.

shows a significant energy lowering. When 1u.c. of LTO is grown on the LCO, interfacial electron transfer occurs and induces the formation of a $CoO_4$ tetrahedral sub-layer. As shown in Fig. 1g, after the interfacial charge transfer the final equilibrium state is a balanced charge state near the interface with the perovskite LaCo²⁺O₃ and LaTi⁴⁺O₃. However, interfacial charge transfer would trigger the phase transition of LCO, then the electrostatic charge at the LCO side could be eliminated. Increasing the LTO thickness, charge transfer might occur between LaTi⁴⁺O₃ at the interface and following LaTi³⁺O₃. Due to there still has electron at Ti $d$-band and the uncharged LCO at the interface, consecutive charge transfer across the LCO/LTO interface and the kinetic

process in LCO where the initial $CoO_4$ sub-layer diffuses into the LCO block occur, allowing the formation of a new $CoO_4$ sub-layer at the interface. This consecutive process results in the ordered distribution of the $CoO_4$ sub-layers inserted into the octahedral matrix, quantitatively evidenced by the experimental observation that each u.c. LTO corresponds to the insertion of one $CoO_4$ layer, i.e. La₃Co₃O₈ and La₂Co₂O₅ forms in the C30/T10 and C30/T15, respectively. Such a quantitative dependence maintains in the C50/T25 with the LCO thickness increased up to 25u.c., indicating that the mechanism of the band-filling modification is 3D indeed. In brief, the 3D band-filling modulation realised in the current work is triggered by the 2D interface

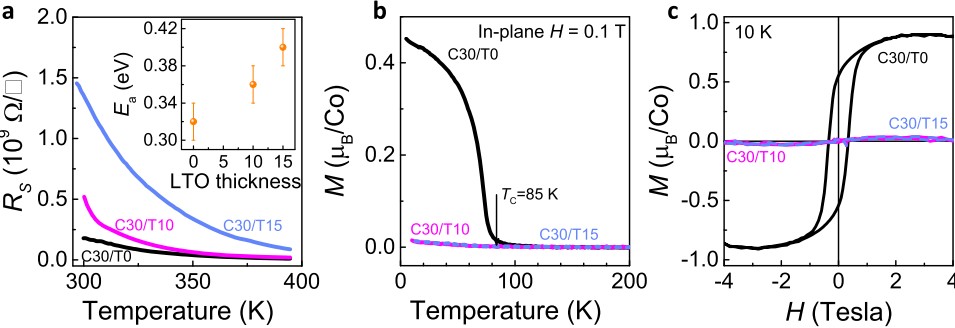

**Fig. 4 Transport and magnetic properties of C30/Tn. a** Sheet resistance of C30/Tn heterostructures as a function of temperature. The inset shows the activation charge gap of C30/Tn. **b** Temperature dependence of magnetisation $M(T)$ curves for C30/Tn under field cooling measured by applying an in-plane magnetic field $H = 0.1$ Tesla. **c** Magnetic hysteresis loops $M(H)$ measured at 10 K for C30/Tn heterostructures. The saturation magnetisation reaches ~0.9 $\mu_B$/Co, evidencing the stoichiometric LCO of the C30/T0[30]. The disappearance of FM ordering in the $La_3Co_3O_8$ and $La_2Co_2O_5$ indicates an AFM ground state as in their bulk form[31, 32].

charge transfer, and mediated by the topotactic phase transition all through the LCO film. We notice a recent report on LCO/LTO superlattices[15], in which the $CoO_4$ tetrahedral layer was not detected. This might be due to the distinct lattice relaxation in the ultra-thin LCO layers (2u.c.) that suppresses the formation of $CoO_4$ tetrahedrons, or the symmetric interfaces at both sides of LCO that result in the random arrangement of $CoO_4$ tetrahedrons.

As characterised by STEM (Fig. 3c), although the LTO layer exhibits the $Ti^{4+}$ oxidation state homogeneously (Supplementary Fig. 2b), it maintains the perovskite structure without any oxygen-excess phases detected. This evidences that the oxygen released from LCO could mainly diffuse through the LTO layer and desorb from the surface during growth. It is worth noting that the LTO layer would be charged within the above picture since the oxygen ions diffused from LCO should leave electrons in LTO when they desorb in the form of molecules. Such a charging effect might be compensated by the STO substrate and capping, which could be considered as electron reservoirs, or by randomly and locally distributed interstitial oxygen in the LTO lattice without being detected by XRD or STEM[44]. Therefore, in the current case, in addition to provide the electrons that trigger the structural destabilisation of LCO, the LTO layer also acts as an oxygen getter layer[20]. Associated with the oxygen out-diffusion, the final state of LCO is indeed the redox-reaction-driven, to the extent of 3D transformation. However, the observed effects cannot be determined by the redox reaction alone due to chemical potential mismatch[45]. We fabricated LCO/oxygen-deficient $LaAlO_3$, and found that the capping layer does induce the redox reaction of LCO but not to the extent of topotactic transitions (Supplementary Fig. 7). We could also directly exclude the current observation is due to the redox reaction of LCO by depositing the amorphous LTO layer[46] (Supplementary Fig. 11). We also fabricated a heterostructure LCO (30u.c.)/STO (5u.c.)/LTO (15u.c.), in which the STO layer effectively blocks the interfacial electron transfer. This sample does not show the topotactic phase transition as observed in the C30/T10 and C30/T15 (Supplementary Fig. 8), indicating that the interfacial charge transfer has a key role in triggering the structural destabilisation and topotactic phase transition of LCO.

It is important to note that the $CoO_4$ tetrahedra induced by the interfacial electron transfer form the 2D network and remain in the layered arrangement during the diffusion away from the interface. This brings the opportunity to design different anion-ordered LCO phases by adjusting the LTO thickness, i.e. the metastable $La_mCo_mO_{3m-1}$ ($m \geq 2$, an integer) can be obtained with arbitrary thickness with the macroscopic distinct electronic

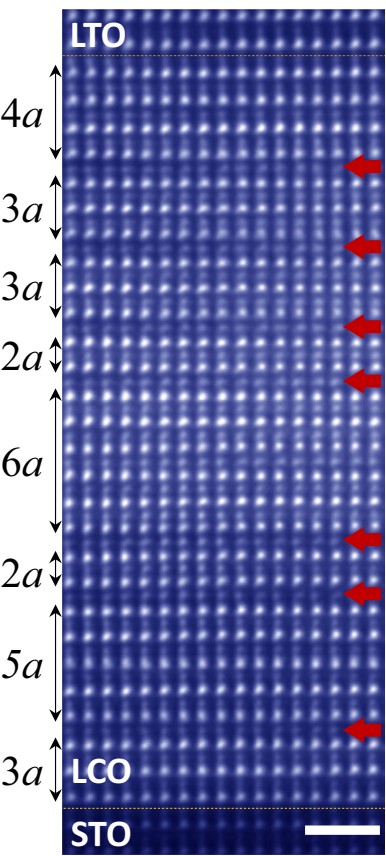

**Fig. 5 Nonuniform distribution of $CoO_4$ layers in C28/T7.** HAADF-STEM image of C28/T7 taken along the STO [110] direction. The scale bar corresponds to 1 nm. The red arrows indicate the $CoO_4$ layers.

and magnetic ground states. Besides the $La_3Co_3O_8$ and $La_2Co_2O_5$ phases identified in XRD (Fig. 3b), we try to obtain the high-ordered phase, $La_4Co_4O_{11}$, by growing the C28/T7 heterostructure. As shown by the HAADF-STEM image in Fig. 5, there are 7 $CoO_4$ sub-layers (the dark stripes indicated by the red arrows) induced by the 7u.c. LTO, appearing as 2D insertions in parallel to the interface. However, their interspacing is not uniform. Different $La_mCo_mO_{3m-1}$ configurations are formed locally without long-range order, resulting in the disappearance of the fractional diffraction peaks in XRD (Fig. 3b). Further fine-tuning of the growth temperature or rate is necessary to control the

kinetic process of the $CoO_4$ sub-layers diffusion, thus the monophased $La_mCo_mO_{3m-1}$ or even the superstructures of the phases with different $m$ can be achieved by design.

In conclusion, we achieve the 3D modulation of the $d$-band-filling of LCO, mediated by the topotactic phase transitions, in the designed LCO/LTO heterostructures. The interfacial electron transfer from LTO to LCO triggers the destabilisation of perovskite LCO, inducing the formation of local lattice configurations with a lowered Co valence. The redistribution of these reduced sub-layers, whose number is determined by the thickness of LTO, mediates the 3D topotactic phase transitions. Our approach can also be extended to other complex oxides, e.g. to achieve the infinite-layer nickelate superconductor through the design of a $Nd_{0.8}Sr_{0.2}NiO_3$/LTO heterostructure[47]. The mechanism proposed in the current work might inspire future works on neuromorphic computing and iontronics.

## Methods

**Thin-film growth, XRD, soft X-ray absorption and magnetic measurements.** Epitaxial heterostructures consisting of LCO and LTO were grown on $TiO_2$-terminated STO (001) substrates using the stoichiometric target of $LaCoO_3$ and $La_2Ti_2O_7$ via pulsed laser deposition (KrF excimer laser, $\lambda = 248$ nm). The laser fluence was 1.5 J/cm². The oxygen partial pressure, repetition rate, and deposition temperature were optimised at 15 Pa, 10 Hz and 670 °C for LCO and at $1 \times 10^{-4}$ Pa, 2 Hz and 670 °C for LTO, respectively. To set symmetric boundary conditions and protect the film from degradation in ambient conditions, we deposited STO capping layers at 500 °C after the growth of LCO and LTO. For reference sample LCO/oxygen-deficient LAO (15u.c.), the LAO was deposited at 670 °C under an oxygen partial pressure of $\sim 1 \times 10^{-4}$ Pa. For LCO/STO(5u.c.)/LTO, the STO was fabricated using the same growth conditions as LTO. The XRD, reciprocal space map and rocking-curve measurements were performed using a Rigaku SmartLab (9 kW) X-ray diffractometer with a Ge (220) × 2 crystal monochromator. The Ti $L$-edge XAS were measured in total electron yield mode at beamline 4B7B of the Beijing Synchrotron Radiation Facility. The Co $L$-edge XAS measurements in total fluorescence yield mode were performed at beamline 11ID-1 at the Canadian Light Source. The spectra were taken at room temperature. The magnetic properties were revealed by the Quantum Design Magnetic Property Measurement System-3, wherein the magnetic field was set to zero in oscillation mode to reduce the residual field of the magnet before measurements.

**STEM image and EELS acquisition.** For STEM image acquisition the cross-sectional STEM specimens were thinned to less than ~30 μm first by using mechanical polishing and then by performing argon ion milling. The ion-beam milling was carried out using PIPS™ (Model 691, Gatan Inc.) with the accelerating voltage of 3.5 kV until a hole was made. Low voltage milling was performed with accelerating voltage of 0.3 kV to remove the surface amorphous layer and to minimise damage. HAADF images were recorded at 300 kV using an aberration-corrected FEI Titan Themis G2 with the convergence semi-angle for imaging 30 mrad and the collection semi-angles snap 39–200 mrad. The EELS data were acquired on a monochromatic Nion-HERME200 aberration-corrected electron microscope operating at 60 kV with a beam current of 20 pA. The beam convergence semi-angle was 20 mrad and the collection semi-angle was 25 mrad. The typical energy resolution (half-width of the full zero-loss peak, ZLP) was 0.7 eV. The typical dwell time was 100–200 ms to achieve a satisfactory signal-to-noise ratio. Gatan Digital Microscopy software and MATLAB were used to process the data. The background of the EELS was fitted using a power-law function and then subtracted, and multiple scattering was removed by a Fourier deconvolution method.

## Data availability

The data that support the findings of this study are available from the corresponding author upon reasonable request.

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

## Acknowledgements

We acknowledge insightful discussions with Erjia Guo. We also thank Dr. Fengmiao Li for TFY mode XAS measurement. The work was supported by the National Key R&D Programme of China (Nos. 2017YFA0303600 & 2016YFA0202300), the National Natural Science Foundation of China (No. 11634016), the Research Programme of Beijing Academy of Quantum Information Sciences (No. Y18G09), and the China Postdoctoral Science Foundation (No. 2018M630219). Y.S., Y.L. and P.G. was supported by the National Key R&D Programme of China (2016YFA0300804), the National Natural Science Foundation of China (Grant Nos. 11974023, 51672007), the Key-Area Research and Development Programme of Guangdong Province (2018B030327001, 2018B010109009).

## Author contributions

J.G. and M.M. conceived the experiments. M.M. and Q.A. grew the samples and performed transport and magnetic measurements. Y.S. and Y.L. performed STEM-EELS measurements under P.G. supervision. M.M. and Z.W. performed x-ray measurements. Z.L. performed DFT calculations. F.Y., X.Z., P.G. and J.G. supervised the project. M.M. wrote the manuscript with contribution from X.Z. and J.G.

## Competing interests

The authors declare no competing interests.
