## [Peer Review File · Nature Communications]

REVIEWER COMMENTS

Reviewer #1 (Remarks to the Author):

In this work, Meng et al employ interfacial charge transfer between LaTiO_3 and LaCoO_3 to induce a topotactic phase transition in LaCoO_3 to $\text{La}_{2/3}\text{Co}_{2/3}\text{O}_5$. This work is very interesting and draws on recent DFT models that predict band alignment and charge transfer across interfaces using the alignment of the O 2p bands between layers. Charge transfer is expected from LaTiO_3 to LaCoO_3 to produce a Ti^{4+} and Co^{2+} formal charge near the interface after electronic reconstruction. References 15 and 16 demonstrate this phenomenon at the LaTiO_3 - LaCoO_3 interface. While there is still much work to be done on these materials, the charge transfer is not sufficiently novel to merit publication in Nature Communications in its own right. Additionally, I have too many concerns regarding the origins of the topotactic phase transition to recommend the work for publication at this time. If the authors are able to address my concerns below, I believe that the paper may be publishable in the future.

1) In Figure 1(f-g), the authors' schematic of the charge transfer from LaTiO_3 to LaCoO_3 is inconsistent with what was proposed in Ref. 12. Per the model put forward in that paper, the O 2p band in LaCoO_3 should also rise as the Fermi level shifts.

2) The authors state that they deposit a SrTiO_3 capping layer on top of the LTO layer to preserve the surface for measurements after atmospheric exposure. However, Fig. S3(a) shows no STO cap on C30/T15. Figure S2(b) shows the Ti oxidation state with position that includes an STO cap, but there is no explanation for how it was acquired. They should show a corresponding image that includes the STO cap.

3) Assuming the cap is there, I have several concerns about the capping layer and its effects on the spectroscopy measurements. For the L-edge XAS shown in Figure 2(d), which was acquired in total fluorescence yield, how did the authors deconvolute the expected Ti^{4+} signal from the STO cap from the Ti^{4+} in the LTO layer? The thicker the cap, the harder it would be to deconvolute. However, if the cap is very thin (a couple unit cells), it will not be a good barrier to protect the surface.

4) The apparent mechanism for the formation of the brownmillerite $\text{La}_{2/3}\text{Co}_{2/3}\text{O}_5$ phase is diffusion of the oxygen all the way through the LTO layer after electrons have been donated. The oxygen then desorbs from the film surface. I am skeptical that LTO would permit that level of oxygen out-diffusion and desorption without scavenging excess oxygen given the stability of the $\text{La}_2\text{Ti}_2\text{O}_7$ phase. In the event that it does, however, the model that the authors have put forward does not conserve charge. The resulting $\text{La}_{2/3}\text{Co}_{2/3}\text{O}_5$ should be a stable phase with Co^{2+} ions that does not depend on electrons donated from LaTiO_3 . Thus, why is the Ti in the 4+ charge state in LaTiO_3 ? In theory the oxygen would move through the LaTiO_3 and then leave behind electrons when it desorbs to return the Ti to a 3+ state.

5) Fig. S3(a) shows that the LTO film is fairly rough on the surface, which is supported by the poor RHEED image at right in Fig. S1(b). Have the authors done spectroscopy on the LTO films to determine oxygen content under their growth conditions? The reference data in Figure 2(d) is taken from the literature, so I am curious what the valence of Ti is for a uniform LTO film from the group. The growth conditions ($P \sim 1 \times 10^{-4} \text{ Pa} = 7.5 \times 10^{-7} \text{ Torr}$) are not too different from the conditions in which STO is deposited in an MBE (low 10^{-6} Torr of O_2) or by high quality via PLD (see Lee et al. Scientific Reports 6, 19941 (2016), DOI: 10.1038/srep19941).

6) The efforts to create control samples with LAO and STO films are admirable and support their conclusions. However, I am still left wondering what is going on in the LaTiO_3 layer as I explained above.

7) The degree of electron diffusion into LaCoO_3 from LaTiO_3 that they postulate is far greater than any other groups have reported in similar systems. For example, Kleibecker et al. PRL 113, 237402 (2014) DOI: 10.1103/PhysRevLett.113.237402 had a rather similar LaFeO_3 - LaTiO_3 interface and saw only 1-2 unit cells of Fe^{2+} due to charge transfer. There are no reports that I'm aware of where charge

transfer has occurred across films that are 15 unit cells thick because the screening length in complex oxides is so small.

8) Fig. S1 should include the time scale. It currently says Time(s) for the horizontal axis, but doesn't provide the actual times.

Reviewer #2 (Remarks to the Author):

The work by Meng et al. reports the finding of restructurings of LaCoO₃ (LCO) thin films when interfaced with LaTiO₃ (LTO). These restructurings are shown to be driven by the (interface) electron transfer from LTO to LCO. Interestingly, the LCO restructuring results in the formation of CoO₄ sublayers in the LCO matrix in a topotactic fashion, and even far from the actual interface. The LCO ferromagnetic (FM) ordering is apparently lost via this interface-triggered phase transition.

The findings of this work are highly interesting and may open a new pathway to the realization of intriguing oxide phases. The writing is sound, albeit somewhat technical at various places. It would be helpful for the reader to render the physics conclusions, impact and outlook of the given results more concise, if possible.

Concrete comments/questions are as follows:

1. It would be helpful to display the differences in the different restructured LCO layers clearer. Fig 3a is small and from inspection the very details of the inserted CoO₄ layers do not become perfectly obvious. A zoom view on the concrete different Co surrounding would improve the presentation.
2. FM order is reported lost from Fig. 4c, but how can the authors detect AFM order? Is the latter one a speculation or proven? The authors should discuss the (possible) magnetic order in some more detail.
3. Is there a possibility to increase the 30 layers substantially? Or in other words, how strongly '3D-like' to the authors expect the reconstruction to be?

This is an interesting new work in the area of oxide heterostructures that deserves highlight recognition. After some refinements in the writing/presentation and addressing of the posed questions, the manuscript can be recommended for publication in Nature Communications.

Reviewer #3 (Remarks to the Author):

Dear Editor,

The manuscript of Meng et al. reports the epitaxial growth and charge reconstruction in a perovskite-based heterostructure of LaCoO₃/LaTiO₃. Besides interfacial charge transfer, the main finding is a reduction of the almost whole LaCoO₃ perovskite to oxygen deficient phases. It is found that the content of the reduction can be controlled by the thickness of the LaTiO₃ thin films, particularly the brownmillerite LaCoO_{2.5} structure is observed in a heterostructure of LaCoO₃ (30 uc)/LaTiO₃ (15 uc) as confirmed by both XRD and STEM measurements. However, both the conductivity and the magnetization of the heterostructures are worse than the bare LaCoO₃ thin films. The results are of potential interest, but there are a number of key questions remain open.

1. The authors explain the finding is due to the charge transfer from LaTiO₃ to LaCoO₃. However, the EELS measurements in Fig.2 show hardly signature of Co²⁺. The XAS in Fig.s2a shows features of Co²⁺, but the analysis in Fig.s2b can not be consistent with the experimental data in

Fig.s2a. I suggest the authors to move the XAS data to the main text and make a further careful analysis. Also, when electrons are transferred to the LCO, the fermi level is expected to be downshift instead of upshift as shown in Fig.1(g).

2. The pure electronic reconstruction at LCO/LTO interface should be of short range nature, see for example Ref.16. So it remains unclear how the electronic reconstruction induces the strong reduction of LCO films. In fact, good LCO growth prefers a high oxygen background pressure, while growing stoichiometric LTO needs a very low oxygen background pressure. To make a compromise, I noticed that the authors grow the LCO at 15 Pa, 670 °C and at 1×10^{-4} Pa, 670 °C for LTO. It is no doubt that the LCO reduction could be due to the film growth of LTO process. The important questions remain are (1) whether the reduction of LCO is due to the annealing in high vacuum (1×10^{-4} Pa) alone or due to the redox reaction or oxygen absorber of the LTO film. Redox reaction will depend on temperature, oxygen pressure, and time, and it has been found that Ti-perovskite could be even more reductive than LAO, see for example Chen et al. Nano letters, 11, 3774, (2011) and 17, 7362 (2017). So control experiments have to be performed to check these two possibilities.

3. What happens to LCO 30 uc when the LTO is thicker than 15 uc? Can you make the LSC 30 uc to the brownmillerite $\text{LaCoO}_{2.5}$ structure by depositing LAO film more than 15 uc?

Response to reviewers' comments

Manuscript Number: NCOMMS-20-33853

We are appreciative of the constructive suggestions and comments from all the reviewers. Accordingly, we have revised the manuscript carefully to improve the quality of the manuscript. Our point-by-point responses are presented below in detail. And we provide a marked version of the revised manuscript in which all the changes are highlighted in yellow.

Reviewer #1

Comment 1: *In this work, Meng et al employ interfacial charge transfer between LaTiO₃ and LaCoO₃ to induce a topotactic phase transition in LaCoO₃ to La₂Co₂O₅. This work is very interesting and draws on recent DFT models that predict band alignment and charge transfer across interfaces using the alignment of the O 2p bands between layers. Charge transfer is expected from LaTiO₃ to LaCoO₃ to produce a Ti⁴⁺ and Co²⁺ formal charge near the interface after electronic reconstruction. References 15 and 16 demonstrate this phenomenon at the LaTiO₃-LaCoO₃ interface. While there is still much work to be done on these materials, the charge transfer is not sufficiently novel to merit publication in Nature Communications in its own right. Additionally, I have too many concerns regarding the origins of the topotactic phase transition to recommend the work for publication at this time.*

Reply: Besides the 2D charge transfer across the interface, the current manuscript emphasizes the accompanied 3D effect mediated by the topotactic transition in the designed LaTiO₃/LaCoO₃ heterostructure. Such a dramatic 3D effect breaks the 2D limit of the charge transfer confined at the interface between ionic oxides. Therefore it is possible to tune the 3D band-filling of oxide films homogeneously and the corresponding functionalities, indicating the potential applications in iontronics *et al.* We have added the above descriptions to the last paragraph of the Introduction Part of the revised manuscript

We totally agree with that the manuscript should clarify the mechanism of the LCO transition. Briefly, it can be describes as an effect of the 2D charge transfer across the LCO/LTO interface, which further triggers the 3D structural destabilization of the perovskite LCO film. We provide the supports of the picture as the following: 1) The interfacial charge transfer due to the difference of the electron affinity [Ref. 12] across the LCO/LTO interface is illustrated [Fig. 1 (e)-(g)]. 2) The structural destabilization of perovskite LCO upon electron transfer is evidenced by the first principle calculations (the first paragraph in the Discussion part of the manuscript with the details given in the Supplemental Material). 3) The macroscopic topotactic transitions of LCO are identified by the XRD and STEM, and found to be dependent on the LTO thickness (Figs. 3, 5, S3, and S10). 4) The additional role of the LTO layer as an oxygen getter to release oxygen from the system is indicated by the homogeneous Ti⁴⁺ oxidation state and its unchanged perovskite structure [Fig. 3 (c) and S2 (b)]. Below please find the point-by-point response to the respective questions raised.

Comment 2: *In Figure 1(f-g), the authors' schematic of the charge transfer from LaTiO₃ to LaCoO₃ is inconsistent with what was proposed in Ref. 12. Per the model put forward in that paper, the O 2p band in LaCoO₃ should also rise as the Fermi level shifts.*

Reply: Thanks for pointing out. We have made the correction in the revised manuscript.

Comment 3: *The authors state that they deposit a SrTiO₃ capping layer on top of the LTO layer to preserve the surface for measurements after atmospheric exposure. However, Fig. S3(a) shows no STO cap on C30/T15. Figure S2(b) shows the Ti oxidation state with position that includes an STO cap, but there is no explanation for how it was acquired. They should show a corresponding image that includes the STO cap.*

Reply: Thanks for the suggestion. We provided the STEM image with an enlarged area in the revised Supplemental Material (Fig. S2a and Fig. S3a) to show the STO capping layer clearly. The Ti oxidation states with position shown in Fig. S2b were determined by the Ti *L* edge peak positions in the spatially resolved STEM-EELS spectra collected along the red dashed line labeled in the STEM image (Fig. S2a). To be clear to readers, we added the corresponding descriptions to the figure captions.

Comment 4: *Assuming the cap is there, I have several concerns about the capping layer and its effects on the spectroscopy measurements. For the L-edge XAS shown in Figure 2(d), which was acquired in total fluorescence yield, how did the authors deconvolute the expected Ti⁴⁺ signal from the STO cap from the Ti⁴⁺ in the LTO layer? The thicker the cap, the harder it would be to deconvolute. However, if the cap is very thin (a couple unit cells), it will not be a good barrier to protect the surface.*

Reply: We thank the reviewer on these insightful comments. We agree with that the Ti⁴⁺ signal from the STO capping layer would contribute to the total XAS results. However, considering the detection depth of XAS (much larger than the capping thickness of 5 u.c.), the Ti ions in the capped LTO layer must make a significant contribution to XAS. The fact that no signals of Ti³⁺-ions were detected in XAS of the C30/T15 indicates the elevated states of Ti in LTO. Additionally, The Ti oxidation states of LTO layer from the C30/T15 in the spatially resolved STEM-EELS (Fig. S2b) has been calculated to be 4+, consistent with the XAS results. Therefore, we conclude that the valence of the Ti ions in the LTO layer can be directly determined as 4+ homogeneously. We have made corresponding revision to the first paragraph on Page 6.

It should be noted that no crystallographic degradation of the perovskite LaTiO₃ was observed in the C30/*T_n* capped with 5 u.c.-thick STO (see Figure S3a for example), indicating that the capping layer indeed protects the heterostructure. This is consistent with the previous report [P. Scheiderer et al., *Adv. Mater.* 30, 1706708 (2018)] claiming that the 5 u.c.-thick capping was enough to protect the LTO film.

Comment 5: *The apparent mechanism for the formation of the brownmillerite La₂Co₂O₅ phase is diffusion of the oxygen all the way through the LTO layer after electrons have been donated. The oxygen then desorbs from the film surface. I am skeptical that LTO would permit that level of oxygen out-diffusion and desorption without scavenging excess oxygen given the stability of the La₂Ti₂O₇ phase. In the event that it does, however, the model that the authors have put forward does not conserve charge. The resulting La₂Co₂O₅ should be a stable phase with Co²⁺ ions that does not depend on electrons donated from LaTiO₃. Thus, why is the Ti in the 4+ charge state in LaTiO₃? In theory the oxygen would move through the LaTiO₃ and then leave behind electrons when it desorbs to return the Ti to a 3+ state.*

Reply: Thanks for the summary of the mechanism we proposed in the manuscript. The LTO layer has been acting as the electron provider and the oxygen getter. And we totally agree with that it is strange for the Ti ions in LTO in the valence of +4. In the revised manuscript, we presented a brief discussion based on the experimental observations: 1) The STO substrate and capping might be acting as electron reservoirs to accommodate excess electrons; 2) The charging of the LTO layer might be partially compensated by a certain amount of interstitial oxygen ions. These interstitial oxygen ions are randomly distributed such that they cannot be detected by XRD or STEM [P. Scheiderer et al., *Adv. Mater.* 30, 1706708 (2018)].

We have added the above discussions to the second paragraph of the Discussion Part in the revised manuscript and wish to motivate further investigations to clarify the characteristics of the LTO layer in the heterostructure.

Comment 6: *Fig. S3(a) shows that the LTO film is fairly rough on the surface, which is supported by the poor RHEED image at right in Fig. S1(b). Have the authors done spectroscopy on the LTO films to determine oxygen content under their growth conditions? The reference data in Figure 2(d) is taken from the literature, so I am curious what the valence of Ti is for a uniform LTO film from the group. The growth conditions ($P \sim 1 \times 10^{-4}$ Pa = 7.5×10^{-7} Torr) are not too different from the conditions in which STO is deposited in an MBE (low 10^{-6} Torr of O₂) or by high quality via PLD (see Lee et al. *Scientific Reports* 6, 19941 (2016), DOI: 10.1038/srep19941).*

Reply: The surface of the LTO layer is atomically smooth as revealed by the sharp interface between the LTO and the STO capping layers (Fig. S3. Note the rough surface in the STEM image corresponds to the damage from STEM sample preparation.). The interface between the LTO and the STO capping layers can be distinguished, as labeled by the dashed lines at the right in Fig. S3a and b. Moreover, we also performed the STEM/EDX mappings of the C30/T15 sample, as shown in Fig. R1. The interface between LTO and STO capping is sharp, with limited Sr/La intermixing. We have added Fig. R1 to the revised Supplemental Material as Fig. S6e.

Fig. R1. STEM/EDX elemental maps of the C30/T15.

To address the questions related to the Ti oxidation state in LTO grown by our group, we have performed additional experiments: growing 50-u.c.-thick LTO on the STO (001) substrate and measuring the *ex-situ* X-ray photoelectron spectroscopy (XPS) of the Ti $2p$ core level. Note that 50 u.c. LTO is much larger than the XPS detecting depth; such that the contribution from the STO substrate can be ignored. After the XPS measurements, we also carried out the structural characterization by XRD. The XRD of the sample [Fig. R2(a)] clearly shows the perovskite structure of LTO. The *in-situ* RHEED patterns of the film [Fig. R2(b)] indicate the flat surface. The Ti $2p_{3/2}$ core level of the LTO film exhibits a mixture of Ti^{3+} and Ti^{4+} with the splitting of $\Delta = 1.2$ eV, consistent with the *in-situ* XPS measurements [P. Scheiderer et al., *Adv. Mater.* 30, 1706708 (2018)]. The Ti^{3+} signal is prevailing despite of the possible oxidation of the surface after a short exposure in the air. For comparison, the Ti XPS spectra of a 50-u.c.-thick BaTiO_3 (BTO) film have also presented in Fig. R2. In BTO, the almost pure Ti^{4+} with narrow FWHM of Ti $2p_{3/2}$ core level has been observed.

Fig. R2. (a) X-ray diffraction data of 50 u.c. LTO grown on STO (001). (b) RHEED pattern of 50 u.c.-LTO after growth. (c) and (d) Ti $2p$ XPS spectra of LTO and BTO.

Comment 7: *The efforts to create control samples with LAO and STO films are admirable and support their conclusions. However, I am still left wondering what is going on in the LaTiO₃ layer as I explained above.*

Reply: As described in *Reply-to-Comment-5* in detail, the LTO layer keeps the perovskite structure while the charging effect might be compensated by the accommodation of some randomly distributed interstitial oxygen or by the STO capping.

Comment 8: *The degree of electron diffusion into LaCoO₃ from LaTiO₃ that they postulate is far greater than any other groups have reported in similar systems. For example, Kleibecker et al. PRL 113, 237402 (2014) DOI: 10.1103/PhysRevLett.113.237402 had a rather similar LaFeO₃-LaTiO₃ interface and saw only 1-2 unit cells of Fe²⁺ due to charge transfer. There are no reports that I'm aware of where charge transfer has occurred across films that are 15 unit cells thick because the screening length in complex oxides is so small.*

Reply: Thanks for pointing out. The 3D effect in the LCO/LTO heterostructures observed in the current work is intriguing. Beyond the 2D charge transfer at the interface as for other oxide interfaces, the structural destabilization of perovskite LCO is triggered [the first paragraph in the Discussion part of the manuscript with the details given in the Supplemental Material], and the 3D topotactic phase transition occurs all through the LCO film with the assistance of the oxygen out-diffusion, resulting in the 3D modification of band-filling.

Comment 9: *Fig. S1 should include the time scale. It currently says Time(s) for the horizontal axis, but doesn't provide the actual times.*

Reply: We have labeled the unit (seconds) of the time scale in the revised Fig. S1 and added the descriptions of the film thickness monitored as the times of RHEED oscillations.

Reviewer #2

Comment 1: *It would be helpful to display the differences in the different restructured LCO layers clearer. Fig 3a is small and from inspection the very details of the inserted CoO₄ layers do not become perfectly obvious. A zoom view on the concrete different Co surrounding would improve the presentation.*

Reply: We have added the zoom in details to Fig. 3a in the revised manuscript.

Comment 2: *FM order is reported lost from Fig. 4c, but how can the authors detect AFM order? Is the latter one a speculation or proven? The authors should discuss the (possible) magnetic order in some more detail.*

Reply: Only with the *M-H* measurements shown in Fig. 4c, we cannot determine the AFM order of the La₂Co₂O₅ or La₃Co₃O₈ films. Instead, by XRD, we first identified the LCO structures in the C30/T10 and C30/T15 samples as La₂Co₂O₅ and La₃Co₃O₈,

respectively. Then, with the previous studies of neutron diffraction on the bulk $\text{La}_2\text{Co}_2\text{O}_5$ and $\text{La}_3\text{Co}_3\text{O}_8$ samples that revealed the AFM orders with Neel temperatures of 301 K [J. Solid State Chem. 141, 411 (1998)] and 35 K [J. Mater. Chem. 8, 2081 (1998)], respectively, we concluded that the C30/T10 and C30/T15 are antiferromagnetic.

This can be understood by analyzing the exchange interactions in the C30/T10 and C30/T15 samples. In a tensile-strained LaCoO_3 film, the Co^{3+} ions may adopt the low-spin/high-spin mixture state [Phys. Rev. B 85, 140404 (2012)]. The ferromagnetic insulating ground state is established via the superexchange interaction between HS and LS Co^{3+} , according to the so-called Goodenough–Kanamori–Anderson rules. However, this interaction is weak and easily vanishes when HS Co^{2+} ($t_{2g}^5 e_g^2$, $S = 3/2$) ions reach a threshold ratio of $\sim 12.5\%$ when the exchange coupling between two Co^{2+} with the half-filled e_g orbital turns out to be strong and antiferromagnetic. Our $M(T)$ and $M(H)$ measurements of the $\text{La}_3\text{Co}_3\text{O}_8$ and $\text{La}_2\text{Co}_2\text{O}_5$ samples [Fig. 4 (b) and (c)] fit the above model well.

To be clear to readers, we added a brief explanation to the captions of Fig. 4 in the revised version.

Comment 3: *Is there a possibility to increase the 30 layers substantially? Or in other words, how strongly '3D-like' to the authors expect the reconstruction to be?*

Reply: Thanks for the intriguing question. The 3D characteristic of the band-filling modification in the LCO/LTO heterostructures is interesting indeed. We have tried to increase the LCO thickness up to 50 u.c. and observed the transitions induced by LTO following the same dependence of the thickness ratio, *i.e.* $\text{La}_2\text{Co}_2\text{O}_5$ in $\text{LCO}_{50}/\text{LTO}_{25}$. Considering that LCO is ferroelastic [Sci. Adv. 5, eaav5050 (2019)], we assume that such 3D-like effect could work within the thickness of the formation of ferroelastic twins that would greatly affect the LCO surface flatness and thus the quality of the heterostructure. This upper limit of LCO thickness is estimated as ~ 90 u.c. when the formation of twin boundary was observed [Phys. Rev. M 3, 074406 (2019)]. To be clear to readers, we added the descriptions of the 3D-like characteristic of the band-filling modification to the revised version.

Reviewer #3

Comment 1: *The manuscript of Meng et al. reports the epitaxial growth and charge reconstruction in a perovskite-based heterostructure of $\text{LaCoO}_3/\text{LaTiO}_3$. Besides interfacial charge transfer, the main finding is a reduction of the almost whole LaCoO_3 perovskite to oxygen deficient phases. It is found that the content of the reduction can be controlled by the thickness of the LaTiO_3 thin films, particularly the brownmillerite $\text{LaCoO}_{2.5}$ structure is observed in a heterostructure of LaCoO_3 (30 uc)/ LaTiO_3 (15 uc) as confirmed by both XRD and STEM measurements. However, both the conductivity and the magnetization of the heterostructures are worse than the bare LaCoO_3 thin films. The results are of potential interest, but there are a number*

of key questions remain open.

Reply: We thank the reviewer for the summary of the current work. Although the resulting phases are more insulating and antiferromagnetic (“worse” than the bare LaCoO_3 film), the 3D characteristic of the mechanism for tuning the LCO structure in the designed heterostructure of C30/Tn is intriguing. Beyond the 2D charge transfer at the interface as for other oxide interfaces, the structural destabilization of perovskite LCO is triggered [the first paragraph in the Discussion part of the manuscript with the details given in the Supplemental Material], and the structural phase transition occurs all through the LCO film with the assistance of the oxygen out-diffusion, making it possible to control the Co *d*-band-filling homogeneously. Moreover, metastable superstructures of $\text{La}_m\text{Co}_m\text{O}_{3m-1}$ could be obtained by tuning the LTO thickness. The mechanism discussed in the current work has potential applications in iontronics *etc.* Below please find the point-by-point response to the respective questions raised.

Comment 2: *The authors explain the finding is due to the charge transfer from LaTiO_3 to LaCoO_3 . However, the EELS measurements in Fig.2 show hardly signature of Co^{2+} . The XAS in Fig.s2a shows features of Co^{2+} , but the analysis in Fig.s2b can not be consistent with the experimental data in Fig.s2a. I suggest the authors to move the XAS data to the main text and make a further careful analysis. Also, when electrons are transferred to the LCO, the fermi level is expected to be downshift instead of upshift as shown in Fig.1(g).*

Reply: We thank the reviewer for the constructive suggestions. In Fig. 2(a)-(c) (old version), the homogeneous L_3/L_2 ratio of Co in the spatially resolved EELS indicated the uniform valence modification all through the 30 u.c. LCO layer in the C30/T15 relative to C30/T0 [*Nat. Mater.* 11, 888 (2012); *Phys. Rev. Lett.* 112, 087202 (2014)]. Moreover, the peak shift of the L_3 -edge in the C30/T15 towards lower energy could also indicate the lowered Co oxidation state [*Phys. Rev. M* 3, 074406 (2019)]. We agree with that the Co *L*-edge XAS that shows the clear splitting is more straightforward to determine the Co valence of +2. Therefore the XAS results were added to the revised Figure 2, with the corresponding descriptions to the second paragraph on Page 5.

Thanks for pointing out the unclear illustration of the Fermi level shift upon the charge transfer. We have revised Fig. 1(g) in the new version.

Comment 3: *The pure electronic reconstruction at LCO/LTO interface should be of short range nature, see for example Ref.16. So it remains unclear how the electronic reconstruction induces the strong reduction of LCO films.*

Reply: Thanks for pointing out the important issue. We totally agree with that the charge transfer itself, driven by the electron affinity difference, should be a 2D process confined at the LCO/LTO interface. With first principle calculations (the first paragraph in the Discussion part of the manuscript with the details given in the Supplemental Material), we have further shown the consequence of the charge transfer, *i.e.* the structure of perovskite LaCoO_3 became unstable with extra electrons from LTO, and CoO_4 tetrahedral layers would be formed associated with oxygen

out-diffusion. Mediated by such a topotactic transition, the effect triggered by the interface charge transfer turns to be 3D. Experimentally we observed the corresponding 3D structural transitions by tuning the thickness LTO. We note that in Ref. 16, the authors have grown a LCO(30)/LTO(4) heterostructure without the tetrahedra layers in LCO detected by STEM. This can be explained by the low thickness of LTO that leads to the low-density tetrahedrons; such that the long-range ordering cannot be formed. To be clear to readers, we added a brief summary to the first paragraph in the Discussion part of the revised manuscript.

Comment 4: *In fact, good LCO growth prefers a high oxygen background pressure, while growing stoichiometric LTO needs a very low oxygen background pressure. To make a compromise, I noticed that the authors grow the LCO at 15 Pa, 670 °C and at 1×10^{-4} Pa, 670 °C for LTO. It is no doubt that the LCO reduction could be due to the film growth of LTO process. The important questions remain are (1) whether the reduction of LCO is due to the annealing in high vacuum (1×10^{-4} Pa) alone or due to the redox reaction or oxygen absorber of the LTO film. Redox reaction will depend on temperature, oxygen pressure, and time, and it has been found that Ti-perovskite could be even more reductive than LAO, see for example Chen et al. Nano letters, 11, 3774, (2011) and 17, 7362 (2017). So control experiments have to be performed to check these two possibilities.*

Reply: Thanks for the insightful suggestions to clarify the mechanism of the LCO reduction. We have grown LCO under high vacuum (1×10^{-6} Pa) and annealed the sample for one hour under the same pressure. As shown in Fig. S9, LCO still maintains the thermodynamically stable perovskite phase. Therefore, in the current work we could exclude that “*the reduction of LCO is due to the annealing in high vacuum*”.

As another control experiment, we have grown C30/T15 heterostructures at different temperatures during the LTO layer deposition (640 to 720 °C). Figure S10 shows the corresponding XRD results. All the three C30/T15 samples are in the brownmillerite $\text{La}_2\text{Co}_2\text{O}_5$ phase. We note that LTO has a narrow growth window — we cannot obtain good crystallinity of LTO out of the above temperature range manifested by the rapid decay of the RHEED patterns during the growth. Within the range, we did not observe any change of the ordering of the octahedron/tetrahedron arrangement. It is indicated that “*the redox reaction or oxygen absorber of the LTO film*” within this range cannot be the key reason for the strong LCO reduction.

We further have performed additional experiments: growing amorphous LTO on the high-quality LCO (30 u.c.) at ambient temperature and under high vacuum (1×10^{-6} Pa). The thickness of the LTO layer is estimated as ~12 nm. Figure R3 shows the XRD result, indicating the enlarged out-of-plane lattice parameter of LCO, *i.e.* the creation of oxygen vacancies, probably due to the oxygen absorption of the amorphous LTO layer that increases the concentration of oxygen vacancies in LCO. However, no topotactic phase transition of LCO has been observed in this reference sample, which can be ascribed to the absence of the interface charge transfer that is indispensable to trigger the topotactic transition of LCO. Therefore, we exclude the

possibility that “the redox reaction or oxygen absorber of the LTO film” alone drives the reduction of LCO. We have added the above results to Fig. S11 in the revised Supplemental Material.

Fig. R3. X-ray diffraction data of LCO(30 u.c.)/amorphous LTO and LCO(30 u.c.).

Comment 5: *What happens to LCO 30 uc when the LTO is thicker than 15 uc?*

Reply: We have grown the C30/T25 sample and the XRD results are shown in Fig. R4. Besides the peaks of the brownmillerite $\text{La}_2\text{Co}_2\text{O}_5$ phase, a peak located round 28° appears. Considering the $\text{La}_2\text{Co}_2\text{O}_5$ phase is the most reduced phase for Co, the whole heterostructure would become unstable by further increasing the LTO thickness. The new peak could be due to a new (unknown) phase / disorders emerging from the LCO or LTO layer. Further investigation and fine tuning will be done to clarify what happens to the heterostructures with LTO thicker than 15 u.c..

Fig. R4. X-ray diffraction data of the LCO(30 u.c.)/LTO(25 u.c.) heterostructure.

Comment 6: *Can you make the LCO 30 uc to the brownmillerite $\text{LaCoO}_{2.5}$ structure by depositing LAO film more than 15 uc?*

Reply: We have grown the LCO(30 u.c.)/LAO(40 u.c.) sample and the XRD results are shown in Fig. S7 (dark green curve). No topotactic phase transition occurs in LCO, *i.e.* the LCO layer maintains the perovskite structure. This reference sample also evidences that, without the charge transfer at the LCO/LTO interface, only the oxygen out-diffusion induced by the capping layer cannot induce the 3D band-filling modulation observed in the current work.

REVIEWER COMMENTS

Reviewer #1 (Remarks to the Author):

While I still have some questions regarding the physical cause of the LCO reduction, I believe that the authors have adequately addressed the concerns that I and the other reviewers raised in the first version. The control samples, modification of the discussion, and improvement of the XAS analysis are sufficient in my opinion to support their conclusions. If the other reviewers agree, I believe that the manuscript can be published in its current form.

Reviewer #2 (Remarks to the Author):

The authors have revised their manuscript according to the various questions/comments from the referees. There are many experimental details to be addressed and the assessment of the different issues appears nontrivial. Still seemingly, the authors provide a great deal of experimental data and insight to support their result and interpretation.

Therefore, the main message and findings appear solid and interesting enough to now warrant support for publication in Nature Communications.

Reviewer #3 (Remarks to the Author):

Dear Editor,

Most of my comments have been answered clearly. The results turn out to be interesting. However, I don't think the author has answered my key concern: it is the redox reaction rather than electronic charge transfer that controls the phenomena observed here. To make my comments clear:

(1) Interfacial electronic charge transfer will mainly occur at the sample surface and/or the interface. So if the electronic effect plays the key role, even there is 3D reconstruction in the LCO layer, there should be some indication on the LTO layer, such as a difference in spatial profile for the Ti^{3+}/Ti^{4+} should be present. Instead, the authors showed/mentioned that the LTO layer exhibits the Ti^{4+} oxidation state homogeneously, so the redox reaction most likely dominates the phenomena observed here and the capping layer of STO may also play an important role as indicated by other referees.

(2) More importantly, in the case of the pure electronic charge transfer, the total amount of transferred charge is generally limited and should be not dependent on the thickness of LTO (or the transition should occur sharply). On page 8, line 185-186, the author mentioned that "the sample with a thicker LTO has a higher content of Co^{2+} " this seems consistent with the redox reaction picture, i.e. the chemical potential drives the reduction of LCO.

(3) The authors mentioned that they have grown amorphous LTO on high-quality LCO (30 u.c.) at ambient temperature and under high vacuum (1×10^{-6} Pa) and only observed limited reduction by XRD. However, the reduction could be much enhanced at high temperatures. The authors are suggested to anneal the amorphous LTO/LCO at the condition similar to the high temperature deposition of LTO, and further checked the crystalline structure of the annealed sample.

There are also two minor issues:

(1) Whether it is "topotactic phase transitions" or redox reduction of LCO? For me the topotactic phase transition often involves other strong reduction agent such as CaH_2 , so it might be more suitable to call it redox reaction.

(2) The first principle calculations in the Discussion part of the manuscript is not convincing.

Shortly, I think the above issues should be clarified before the publication of the manuscript.

Response to reviewers' comments

Manuscript Number: NCOMMS-20-33853A

We thank all the reviewers for their careful reading and helpful comments of our manuscript. The fact that Reviewer #1 and #2 find our revision being well improved and acceptable for publication in *Nature Communications* is rewarding. Reviewer #3 further raises some issues about the mechanism of the transition of LCO. Our point-by-point responses to the comments are presented below in detail.

Reviewer #3

Comment 1: *Interfacial electronic charge transfer will mainly occur at the sample surface and/or the interface. So if the electronic effect plays the key role, even there is 3D reconstruction in the LCO layer, there should be some indication on the LTO layer, such as a difference in spatial profile for the Ti³⁺/Ti⁴⁺ should be present. Instead, the authors showed/mentioned that the LTO layer exhibits the Ti⁴⁺ oxidation state homogeneously, so the redox reaction most likely dominate the phenomena observed here and the capping layer of STO may also play an important role as indicated by other referees.*

Reply: We totally agree with that the interfacial charge transfer between complex oxides should be quasi-two-dimensional (2D). This 2D charge transfer at the LCO/LTO interface triggers the structural transition in LCO, which is associated with the release of oxygen ions. The net charge on the LCO side is canceled and the electric field that confines the charge transfer within the 2D region is changed. Consequently, the LTO layers away from the interface would provide more electrons diffusing towards the interface. In brief, the initial 2D charge transfer at LCO/LTO triggers the structural phase transition, further changing the screening electric field, and resulting in the homogeneous valence states across the films at both sides. The key role of the initial 2D charge transfer at LCO/LTO (with the density as high as $\sim 6.6 \times 10^{-14} \text{ cm}^{-2}$) in triggering the structural phase transition has been evidenced by the reference samples of LCO/STO or LCO/LAO (Supplemental Material, Fig. S7), in which the mismatch of the oxygen concentration does introduce some oxygen vacancies (reduce) to the perovskite LCO, but the amount is far less than the extent to induce the structural phase transition.

It is worth noting that the Ti ions in LTO are in the valence of 4+, instead of 3+ as in the perovskite structure. We have presented a brief discussion based on the experimental observations: 1) The STO substrate and capping might be acting as electron reservoirs to accommodate excess electrons; 2) The charging of the LTO layer might be partially compensated by a certain amount of interstitial oxygen ions. In this sense, it might be important to cap the heterostructures with the amorphous layer, but not limited to STO specifically. For example, it was reported that LaNiO₃ could also be effective in compensating the excess charge of the underlying layer [Ref. 16].

To be clear to readers, we have made corresponding revisions to the first and second paragraphs of the Discussion part (Page 9&10) of the revised manuscript, and wish to motivate further investigations to clarify the in-depth mechanism.

Comment 2: *More importantly, In the case of the pure electronic charge transfer, the total amount of transferred charge is generally limited and should be not dependent on the thickness of LTO (or the transition should occur sharply). On page 8, line 185-186, the author mentioned that “the sample with a thicker LTO has a higher content of Co²⁺” this seems consistent with the redox reaction picture, i.e. the chemical potential drives the reduction of LCO.*

Reply: Thanks for pointing out the important issue. The initial charge transfer at the LCO/LTO interface is indeed 2D. But since the structural phase transition of LCO is triggered, the valence state of Co varies, canceling the net charge on the LCO side and changing the electric field at the interface that confines the charge transfer within 2D. As detailed in the Reply-to-Comment#1, the LTO layers away from the interface would provide more electrons diffusing towards the interface, resulting in the scenario that the total amount of the transferred charge is proportional to the LTO thickness, consistent with the experimental observations. To clarify the issue, we have added the above descriptions to the first paragraph of the Discussion part (Page 9) of the revised manuscript.

In addition, we exclude the possibility of that “*the chemical potential drives the reduction of LCO*” as the only mechanism by three facts: (1) We have compared the formation and migration energy of oxygen vacancies in different oxides (Table R1). Since the structural phase transition of the LCO film is associated with the mobile oxygen ions diffusing out of the LCO through the capping layers (LTO in the current work), the upper layers with lower migration energy would lead to stronger reduction of LCO. As listed in Table R1, STO should have caused the strongest redox reaction among STO, LAO, and LTO. But experimentally, by capping LAO or STO layers, some oxygen vacancies are introduced to the perovskite LCO, but the amount is far less than the extent to induce the structural phase transition (Fig. S7 in Supplemental Material). (2) Similarly, by growing LCO films in the redox conditions, oxygen vacancies can be created but fail to obtain the La₃Co₃O₈ or La₂Co₂O₅ other than the perovskite phase. (3) The reference samples (Supplemental Material, Fig. S8) indicates that without the interfacial charge transfer, the structural transition of LCO will not occur.

	Migration Energy (eV)	Formation Energy (eV)
LaCoO ₃ [1]	0.7	2.2-3.6
LaTiO ₃ [2]	1.6	5.09-6.2
SrTiO ₃ [3]	0.39-0.49	6.22-6.94
LaAlO ₃ [3]	0.63	4.2-6.5

Table R1. Oxygen vacancy migration energy and formation energy (eV). Ref [1]: Phys. Rev. B 82, 115435 (2010); Ref [2]: J. Phys. Chem. C 118, 28776 (2014); Ref [3]: Adv. Mater. 30, 1705904 (2018);

Comment 3: *The authors mentioned that they have grown amorphous LTO on high-quality LCO (30 u.c.) at ambient temperature and under high vacuum (1×10^{-6}*

Pa) and only observed limited reduction by XRD. However, the reduction could be much enhanced at high temperatures. The authors are suggested to anneal the amorphous LTO/LCO at the condition similar to the high temperature deposition of LTO, and further checked the crystalline structure of the annealed sample.

Reply: We have annealed the LCO/amorphous LTO sample at 670 °C for one hour. As shown in Fig. R1, LCO maintains the perovskite structure after annealing, evidencing that, without the triggering of the interfacial charge transfer, the treatment in a reductive environments alone cannot induce the structural phase transition of LCO. We have added above results to the revised Supplemental Material (Fig. S11).

Figure R1. X-ray diffraction data of LCO(30 u.c.), as-grown and annealed LCO(30 u.c.)/amorphous LTO.

Comment 4: *Whether it is “topotactic phase transitions” or redox reduction of LCO? For me the topotactic phase transition often involves other strong reduction agent such as CaH₂, so it might be more suitable to call it redox reaction.*

Reply: It is true that, in the current work, the phase transition of LCO is indeed a redox reaction since it is accompanied by the loss of oxygen and the change of Co oxidation state. And the crystalline orientation of the lattice is maintained in the phase transition, satisfying the definition of topotactic phase transition [Nat. Mater. 12, 1057 (2013)]. Therefore, to emphasize the ordering of LCO after phase transition, we are inclined to use “topotactic phase transitions”. To be clear to the readers, we have added the descriptions about redox reaction to the Discussion part of the revised manuscript.

Comment 5: *The first principle calculations in the Discussion part of the manuscript is not convincing.*

Reply: The results of the first principle calculations just provide the supplemental demonstration for the structural destabilization of LCO upon the electron transfer, which is mainly based on the experimental observations, *i.e.* the interfacial charge transfer does trigger the structural phase transition in LCO (see Reply-to-Comment 1 for details).

REVIEWERS' COMMENTS

Reviewer #3 (Remarks to the Author):

Dear Editor,

I agree that the additional annealing measurement further supports the authors' claims. The answers to my final questions as well as the revision to the main text are also acceptable. This is a very interesting finding and I recommend its publication without further delay.